# A Critical Analysis of TKR In Vitro Wear Tests Considering Predicted Knee Joint Loads

**DOI:** 10.3390/ma12101597

**Published:** 2019-05-15

**Authors:** Saverio Affatato, Alessandro Ruggiero

**Affiliations:** 1Laboratorio di Tecnologia Medica, IRCCS Istituto Ortopedico Rizzoli, 40136 Bologna, Italy; 2Department of Industrial Engineering, University of Salerno, 84084 Fisciano, Italy; ruggiero@unisa.it

**Keywords:** total knee replacement, musculoskeletal multibody model, knee joint forces, in vitro wear testing

## Abstract

Detailed knowledge about loading of the knee joint is essential for preclinical testing of total knee replacement. Direct measurement of joint reaction forces is generally not feasible in a clinical setting; non-invasive methods based on musculoskeletal modelling should therefore be considered as a valid alternative to the standards guidelines. The aim of this paper is to investigate the possibility of using knee joint forces calculated through musculoskeletal modelling software for developing an in vitro wear assessment protocol by using a knee wear simulator. In particular, in this work we preliminarily show a comparison of the predicted knee joint forces (in silico) during the gait with those obtained from the ISO 14243-1/3 and with those measured in vivo by other authors. Subsequently, we compare the wear results obtained from a knee wear joint simulator loaded by calculated forces in correspondence to the “normal gait” kinematics with those obtained in correspondence to the loads imposed by the ISO. The obtained results show that even if the predicted load profiles are not totally in good agreement with the loads deriving from ISO standards and from in vivo measurements, they can be useful for in vitro wear tests, since the results obtained from the simulator in terms of wear are in agreement with the literature data.

## 1. Introduction

Total knee replacement (TKR) has become a common surgical procedure to alleviate pain and increase functional mobility in diseased or traumatized knee joints [1]. A major limiting factor to the service life of TKRs remains the wear of the polyethylene tibial plate. Preclinical endurance testing has become a standard procedure to predict the mechanical performance of new devices during implant development; on the other hand, these kinds of mechanical wear tests have a long duration and they are very expensive [2]. Ideally, the preclinical endurance testing should reproduce on the artificial implants the in vivo conditions of load and motion experimented by the human joint in order to assess the wear of prosthetic components and give information about materials and design optimization for prosthetic implants. The research in this field has been going on for many years [3,4] and is still strongly active in order to further increase the service life of TKRs, as evidenced by the many studies [5,6,7]. At the state of the art, two simulation concepts for the knee joint are available and defined by the International Organization for Standardization (ISO) 14243-1/3 standards: the force control (FC) and the displacement control (DC) [8,9]. The level walking is the sole activity of daily living that is represented for testing. Recently, significant differences in the features of wear and surface finish between knee replacements subjected to in vitro simulation and implants obtained from explants have been noticed [7,10]. These discrepancies may be due to differences between the kinematic and dynamic profile impressed on the in vivo joint and the one defined by the ISO. Therefore, the research is exploring more accurate methods of analysis that allow for movements and loads acting on knee joint to be obtained. Some authors have focused on the analysis of loads acting on the knee joint for different activities, such as level walking, climbing stairs, or running, in order to estimate a more realistic load profile and thereby, obtain a more realistic behavior of the artificial knee joints during wear simulations [11,12,13,14,15]. Bergman et al. [14] focused on knee joint kinetics, using a transducer, installed inside the implanted prosthesis on the human subject, which measures the forces and moments acting on knee during different in vivo motion conditions. Battaglia et al. [11] used a different approach in which the kinematics data of the knee joint (such as the angle of flexion/extension, the internal/external rotation, and the anterior/posterior translation) are obtained by fluoroscopic analysis. Although the previously described techniques allow an accurate measurement for the analyzed variables, some of them are extremely invasive. For this reason, some studies [16,17,18,19,20] are addressing the issue of predicting joint loads from measured gait patterns using software for the simulation of the musculoskeletal system, in which joint reaction forces are computed from kinematic data and ground reaction force collected using special motion capture tools in gait analysis laboratories [21,22,23,24,25]. Typically, the musculoskeletal system is assumed to be a rigid-body system and combines the principles of the inverse dynamics with different algorithms to define the muscle recruitment and compute loads acting on different joints of the human body. The purpose of inverse dynamics is to estimate forces and moments that cause a particular motion, and the model must have reasonable representations of the muscle geometry and the recruitment pattern of the muscles. Computational models that can represent the musculoskeletal systems of each patient are today used to explore different treatment options and optimize clinical outcomes [26]. Currently, multibody software is becoming of fundamental importance both in the biomedical and ergonomic field. Its use for practical application, adapted to the individual patient, holds significant promise and enormous challenge because direct measurement of joint reaction forces is generally not feasible in a clinical setting. In silico methods based on musculoskeletal modelling simulations are powerful tools that allow biomechanical investigations and predictions of muscle forces not accessible with experiments; this allows us to develop subject-specific models able to simultaneously predict muscle, ligament, and knee joint contact forces, which may be used to test orthopaedic implants with patient-specific load configuration [27,28,29]. The multibody software provides an excellent and non-invasive solution that could be used on any patient. On the contrary, this procedure does not allow a direct measurement of the forces acting on the joints; therefore, in order to confirm the reliability of the load profile processed it is necessary to compare elaborated output with measured ones. 

In the context of the proposed limitations, we investigated the ability of a multibody model to effectively predict knee joint forces for in vitro wear tests. To do this, we split the research in two main parts: the first part was devoted to a novel comparison method of the calculated loads, obtained with three different gait data, with real in vivo conditions reported in the literature [14] and with the ones reported in the ISO 14243-3 standard and actually used in the in vitro testing. In the second step we used the in silico knee loads obtained from normal gait data as knee simulator loads for the in vitro wear simulation, in order to acquire wear differences on the prostheses components with respect to the results obtained with the standard ISO guidelines regarding the TKR loading on simulators.

## 2. Materials and Methods

### 2.1. Anybody System

In this study, loads acting on knee joint were obtained in silico by using the AnyBody Modelling System^TM^ (AMS) computer software (AnyBody Technology A/S, Aalborg, Denmark) [25], accounting for three different gait patterns [30]. The lower limb model considered consisted of an 18 degrees of freedom (DoF) kinematic chain (lower limbs) containing seven segments, the pelvis, and three segments in each leg: thigh, shank, and foot. All segments of the biomechanical system are modelled as rigid bodies connected by ideal kinematic pairs, neglecting effects such as the wobbling masses of the soft tissues. The model contained 70 muscle units based on a Hill type model and was geometrically modelled as elastic strings spanning between two or more points, and in some cases wrapping over rigid obstacles. In the current study, the muscular recruitment was calculated choosing the polynomial criteria, with *p* = 2, since it is established in the literature as predicting reasonable muscle activation patterns for the type of analyzed trial [17,30]. 

The software inputs are kinematic data and ground reaction force, according to the inverse dynamic solution approach. In order to have a general comparison between the simulated results and the loads patterns reported in the ISO 14243-1 and obtained by Bergmann [17], we used three different sets of input data: the first one was created with ground reaction forces and kinematic data from Vaughan et al. [21] and the other two were from the Gait Lab of Aalborg University: Gait Normal (simulating a normal walking movement) and Gait Fast (simulating a faster walking movement).

### 2.2. Validation Anybody Framework

The debate about validation of multibody musculoskeletal models is still open [30], especially in biomechanics. Direct comparison between a model output of interest with an experimental measurement of the same quantity, can be inherently difficult, costly, and ethically problematic to perform. Thus, in this study, in order to assess the difference of the calculated forces predicted through the AnyBody Modelling System^TM^, we proceeded with the following steps:Definition of a range of variability of the loads acting on the knee measured in vivo by analyzing the results obtained in [14].Comparison between the simulated loads and those provided by the reference standard ISO 14243.

Regarding the validation of multibody musculoskeletal models [24], we used a validation metric proposed by Geers [30] which was selected as a possible metric for the comparison of calculated and measured transient response histories [31]. The method defines the calculation of two indexes: a magnitude error (*M*), which is insensitive to inaccuracies on the abscissa, and a phase error (*P*) that does not depend on the amplitude of the curves, as follows:(1)$$M=\sqrt{\frac{v_{cc}}{v_{mm}}-1}$$
(2)$$P=\frac{1}{\pi }\cos^{-1}\frac{v_{mc}}{\sqrt{v_{cc}v_{mm}}}$$

In which:(3)$$v_{mm}=\frac{1}{t_{2}-t_{1}}\overset{t_{2}}{\underset{t_{1}}{\int }}m{\left(t\right)}^{2}dx$$
(4)$$v_{cc}=\frac{1}{t_{2}-t_{1}}\overset{t_{2}}{\underset{t_{1}}{\int }}c{\left(t\right)}^{2}dx$$
(5)$$v_{mc}=\frac{1}{t_{2}-t_{1}}\overset{t_{2}}{\underset{t_{1}}{\int }}m\left(t\right)c\left(t\right)\;dx$$
where *c*(*t*) and *m*(*t*) are respectively the computed loads (in time, during one gait cycle) and the compared function in time. The metric *M* and *P* are designed to give zero when the curves are identical. 

Since the proposed method represents a novel approach, there is no exact discussion as to what is an acceptable level of error. The acceptable values for the error factors will be connected to the specific application to evaluate, and the to the intent of the evaluation. As a general guideline, it is possible to assume that when the *M* and *P* coefficients are below 0.2 it reveals a really good agreement between the compared curves; values around 0.2–0.4 show a fair agreement, while exceeding the value of 0.4 the agreement can be considered poor. The metric, also, provides a combined error (*C*) to produce a single global value for the comparison. With reference to this approach, the data obtained for the three gait patterns were processed with the average loads provided by Bergmann et al. [14] and with the considered standard ISO 14243-3.

### 2.3. Comparison with In Vivo Measuremenst and with ISO14243-1

The comparison with ISO14243-1 Standards is proposed in the tibial reference frame. Forces and moments are reported in *N* and *Nm* respectively.

### 2.4. Knee Wear Simulator

The experimental protocol for the simulator, including kinematics and load data, was implemented on a DC knee wear simulator (Shore Western Mfg., Monrovia, USA), which is a “three-plus-one” station machine by re-programming the control unit of the apparatus. This simulator is used extensively to assess wear from knee joint prostheses [7,11]. The test was performed on three mobile ultra-high-molecular-polyethylene (UHMWPE) coupled with metallic femoral components (Size 2, Genus mobile bearings, Ala ORTHO S.r.l., Milan, Italy) for 1,000,000 cycles. Axial load was applied vertically (perpendicular to the tibial tray), oscillating between 168 and 2600 N following the calculated profile. The applied kinematics was derived from the displacement control simulator [7]. In particular, the flexion/extension angle oscillating between 0° (neutral) and 60° (flexion) was synchronously with the load; the anterior/posterior translation oscillating between −6.0 mm (neutral) and 6.0 mm (posterior), and the intra/extra-rotation oscillating between −2.0° (extra-rotation) and 6.0° (intra-rotation). The test duration was set at 1 million cycles, in agreement to other studies [32] under a frequency of 1.0 ± 0.1 Hz. Each station was filled with distilled water.

### 2.5. Gravimetric, Roughness, and Comparison Methodology

Gravimetric wear of the tibial specimens was assessed at 0.5 Mc and at 1.0 Mc. Mass loss was measured using a microbalance (SARTORIUS AG, Göttingen, Germany) with an uncertainty of 0.01 mg and an accuracy of 0.01 mg. Each mass measurement was repeated three times and the mean values were used. After 1 Mc, all components were visually examined for any other damage. Surface roughness of all femoral and UHMWPE total knee prostheses (TKP) were investigated using a Hommel Tester T8000 (Hommel Tester T8000; Hommel Werke, Schwenningen, Germany) and the SensoFar instruments (SensoFar Metrology, Terrassa, Spain). Roughness was measured on both tibial and femoral metallic surfaces. Before measurements, all components were cleaned with acetone (A.C.E.F., Piacenza, Italy). Ra, Rt, and Rsk were quantified as roughness indicators in agreement with previous studies [6,33]. Ra is the arithmetic average value of the deviations of the roughness profile filtered from the mean line into the sampling length (DIN 4768, ISO 4287), while Rsk is the skewness of the profile and indicates the symmetry of the profile. The combination of these two parameters allows us to better characterize the superficial changes of the prosthetic specimens. A cut-off of 0.25 mm and a sampling length of 1.5 mm were used. Roughness measurements were carried out at the beginning and at the end of the tests. 

After the tests, the surfaces of the specimens were also examined for scratches or damage caused by third-body wear. Scanning electron microscope (SEM, Zeiss Leo 1530, Cambridge, UK) operating at 20 kV and in “variable vacuum” mode (VPSE) was used to analyze all the knee components.

The output of the model obtained from the Anybody software, in terms of axial load components acting on the knee joint, was used to reprogram the knee simulator with the aim to compare the new wear results with previously obtained in vitro wear results [34]. 

## 3. Results

All the specimens tested completed the planned one million of cycles. 

### 3.1. Anybody Modelling Results

The first comparison was done for the resultant force and moment acting on the knee joint. Figure 1 shows, according to the qualitative validation, the range of resultant force and moment provided by instrumented knee prosthesis (identified with the tag “Bergmann’s Range”) versus resultant force and moment simulated by the AnyBody Modelling System for the analyzed gait patterns: “Gait Vaughan”, “Gait Normal”, and “Gait Fast”. 

Figure 2 represents a comparison between in silico and the ISO 1423-3 values.

The resultant simulated force is shown in Figure 3. It is possible to observe a good correlation with experimental measures for both Magnitude and Phase errors (M and P coefficient are always less than 0.2). Regarding the resultant moment (Figure 3), the magnitude error for “Gait Normal” and “Gait Fast” is higher than 0.4, indicating that simulated resultant moment is greater.

The results of the quantitative comparison are proposed in Table 1, Table 2 and Table 3. In particular, in Table 1 it is possible to observe the metric applied for force and moment components. 

Table 2 shows the applied for force and moment components.

Table 3 shows the comparison between in silico vs. ISO 14243.

### 3.2. Simulator Results

The calculation of the dynamic loads acting on the knee shows a great prevalence in magnitude of the axial components during the whole gait, justifying focus on the in vitro simulations on the effect of this load component on the wear phenomena. With this in mind the authors investigated the differences of both the calculated axial force component and the femoral/extension movement with respect the effective displacement law of the simulator after the new programming. The patterns of variations over the waveforms of the axial load (calculated in correspondence of normal gait kinematics, Figure 2b) and the femoral/extension movement (Figure 4) versus the time were then considered in the in vitro simulations. 

### 3.3. Wear Simulator Results

At the end of the test the measured average mass loss of the three menisci is shown in Figure 5. The menisci showed a linear weight loss (R^2^ = 0.98) over the intra-test assessment intervals, which resulted in a cumulative average mass loss of 13.5 ± 3.1 mg. Both medial and lateral menisci showed a similar wear pattern.

The worn UHMWPE inserts deriving from the ISO and multibody simulation were analyzed in the most worn areas. The analyses were performed using the SEM in order to assess the pits and scratches on the menisci surfaces. The area and extent of the focal scar damage observed on the menisci components used under ISO waveforms were different from the menisci components used under in silico simulation. In fact, in Figure 6a,b, it is possible to observe, on the meniscus that run under the ISO waveforms, scratches pattern along the femoral sliding (anterior/posterior (AP) direction). Pits of various dimensions were observed on the articulating surface, and craters were also detected on the lateral compartment. These scratches were probably generated during the wear tests by third-body particles produced in turn by the polishing and entrapment phenomena between the bearing surfaces. A small amount of burnishing and scratches were visible along the AP direction of the menisci that run under in silico multibody waveforms (Figure 6c,d).

## 4. Discussion

A complex morphometric relation characterizes the knee and currently at the authors’ knowledge, there are no studies about the combined effect of patient weight and load applied on the knee. An in vivo performance of knee prostheses, according to the relation between patient body mass index (BMI) and load applied, showed high variability of load according to the anthropometric features specific to each subject [6,12,13,15,16]. These studies, performed on different people with similar weight and height, showed different load configurations due to the different ways in which anyone runs a specific movement.

The main aim of this paper was to investigate the possibility of using calculated (in silico) knee joint forces through musculoskeletal modelling software for developing an in vitro wear assessment protocol to be used in a knee wear simulator. In particular, in this work we preliminarily showed the results of a quantitative comparison of the calculated knee joint forces during the gait with those obtained from the ISO 14243 and with those measured in vivo by other authors. Subsequently, we compared the wear results obtained from a knee wear joint simulator loaded by calculated forces (axial load) in correspondence to the “normal gait” kinematics with those obtained in correspondence to the loads imposed by the ISO.

Our study focused on outlining a consistent protocol for DC knee wear simulation under the standard [34,35] and “high-demanding-activity” conditions [11,12], integrating a musculoskeletal multibody loads calculation procedure using different load profiles calculated from real life activities kinematical data in order to predict and reduce knee wear phenomena. Furthermore, the knowledge of the more realistic load profiles acting on the joint during walking gives the opportunity to investigate in an accurate way on the unsteady lubricating phenomena acting in the tribosystems by using novel lubrication models [36,37].

Originality of the present work can be seen in the new experimental protocol that takes into consideration an innovative combination of three standard measuring techniques. The results of this study might provide an evident advantage from a tribological point of view; the knee wear simulator has shown to reproduce “accurately” the obtained profiles, beyond the ISO 14243 standard waveforms.

The obtained results show that even if the in silico load profiles are not in totally good agreement with the loads deriving from the ISO standards and from the in vivo measurements, from a wear point of view, they could give important information. In fact, from Figure 5 it is possible to observe that all the menisci components exhibit a slight but continuous increase in the weight loss. These results compares very well with the results obtained from previous wear tests in which metal-on-polyethylene prostheses were run on the same knee simulator [33,38]. Vice versa, the results shown in Figure 3 simulate a good correlation with the experimental measures for both magnitude and phase errors, especially in the case of gait Vaughan. It should be also specified that for the measured load profiles large differences have been observed. Bergmann himself highlights the potential differences within the group of total knee replacement (TKR) patients and states that, even though the joint loads were collected from the largest group of subjects with instrumented knee implants currently available, the data would be different if more subjects were included in the study. 

Obviously, the present study has some limitations due to the lack of analyzed subjects available for direct comparison between the compared approaches. Moreover, all the subjects considered in the motion captures analyses were healthy individuals and as such would have a different gait compared to TKR patients. In addition, TKR may have altered gait patterns due to arthroplasty or replacement joints. Although this validation may provide some evidence that knee joint reaction force is predicted as expected and is close enough to the real in vivo conditions, it does not guarantee that the model correctly predicted the actual quantity of interest.

Loading on knee joint primarily depends on the physical activity and body weight (BW). The high variability of loads, according to the anthropometric features specific to each subject, is observed from several studies [6,12,13,15,16]. These studies also point out that different people, with similar weight and height, show different load configurations due to their different ways to run specific movements. From a clinical point of view, given the high variability of loads, direct measurement of joint reaction forces is generally not feasible in a clinical setting; there is a great interest to develop a new in silico approach, based on musculoskeletal modelling [39], in order to assess the loads acting on the knee joint and define which loads are appropriate for mechanical tests of TKR. The results of this study might provide advantages from a tribological point of view; we investigated the ability of a multibody model to effectively predict knee joint forces for in vitro wear tests. These results offer the possibility of extending the range of such in silico approaches and enhancing the reliability of the relevant conditions imposed on wear simulators. The next step, in fact, might be using this new protocol in these simulators to compare the corresponding results with those obtained.

## 5. Conclusions

We can conclude that the gait models obtained by using musculoskeletal multibody models are quite close to the literature data derived from Bergmann’s experimental results when allowing for patient-to-patient variation and variation within patients’ own gait cycle. It was stressed that the load profile provided by the ISO guidelines is not representative of those that are the “real conditions” to which the knee joint is subjected daily. In fact, ISO 14243-1/3 refer only to level walking and does not take into account activities such as climbing stairs or sitting, which are, in any case, activities that are normally performed in the daily living of TKR patients. Currently, wear tests on knee prostheses are performed using expensive knee wear simulators. This work is meant as a contribution towards the definition of a general in vitro wear assessment protocol, based on the in silico load determination by using musculoskeletal modelling systems, which reproduces useful and quick results even with patient-specific load conditions, for a better pre-clinical prostheses testing.

## Figures and Tables

**Figure 1 materials-12-01597-f001:**
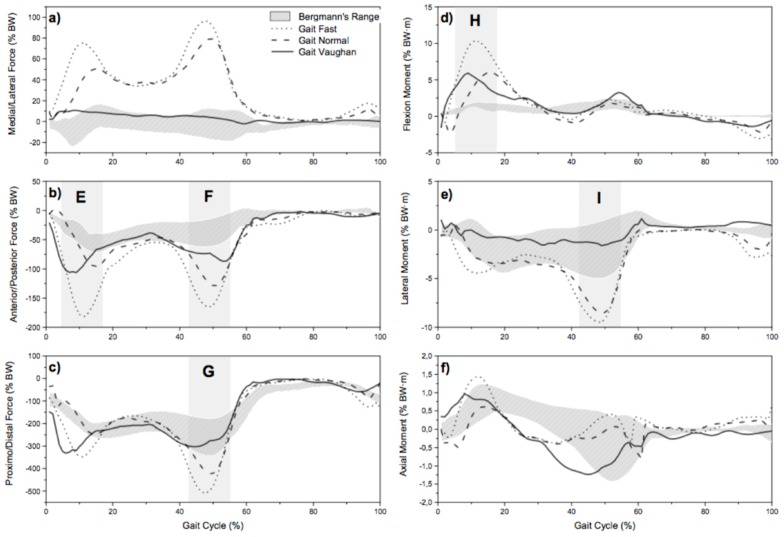
Comparison between in silico: medial/lateral force (**a**), anterior/posterior force (**b**), proximal/distal force (**c**), flexion moment (**d**), lateral moment (**e**), and axial moment (**f**) from three gait types and the in vivo values obtained by Bergman [14].

**Figure 2 materials-12-01597-f002:**
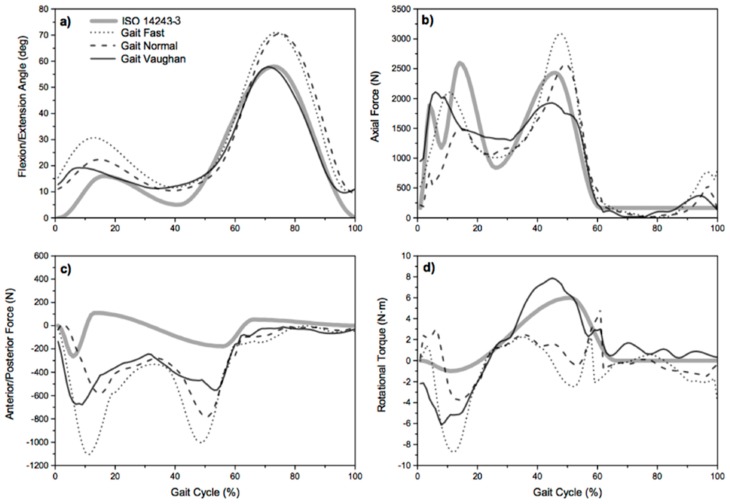
Comparison between in silico: flexion/extension angle (**a**), axial force (**b**), anterior/posterior force (**c**), and rotational torque (**d**) from three gait types and the ISO 1423-3 values.

**Figure 3 materials-12-01597-f003:**
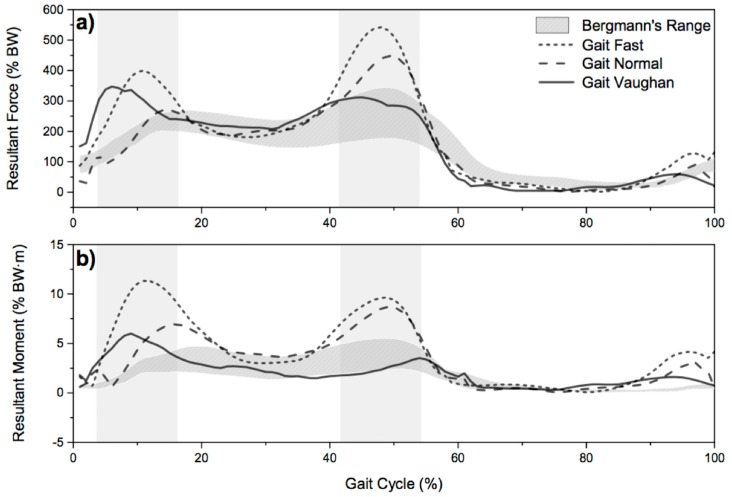
Comparison between in silico resultant force (**a**) and resultant moment (**b**) from three gait types and the in vivo values obtained by Bergman [14].

**Figure 4 materials-12-01597-f004:**
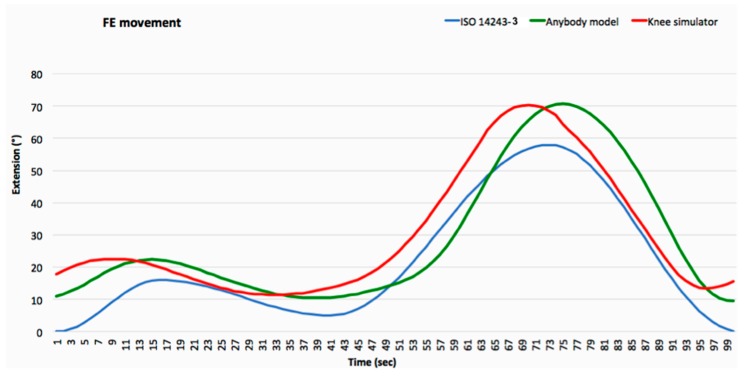
Representation of the femoral/extension movement of the Anybody model of the ISO standard 14243-3 and knee simulator.

**Figure 5 materials-12-01597-f005:**
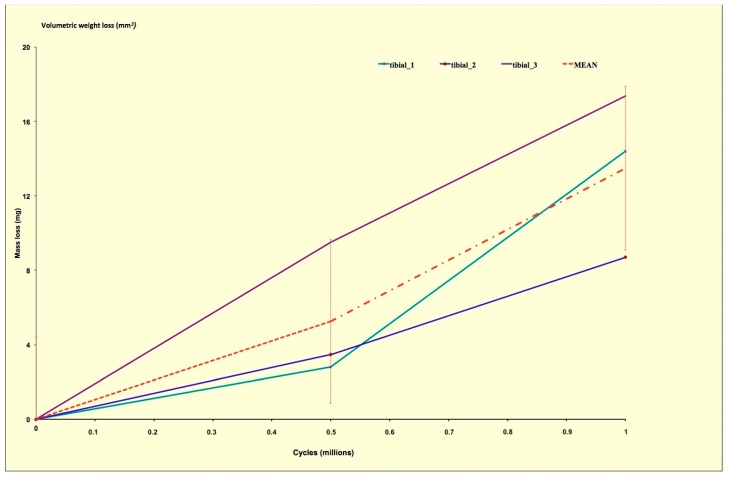
Measured average mass loss of the three menisci tested in this study.

**Figure 6 materials-12-01597-f006:**
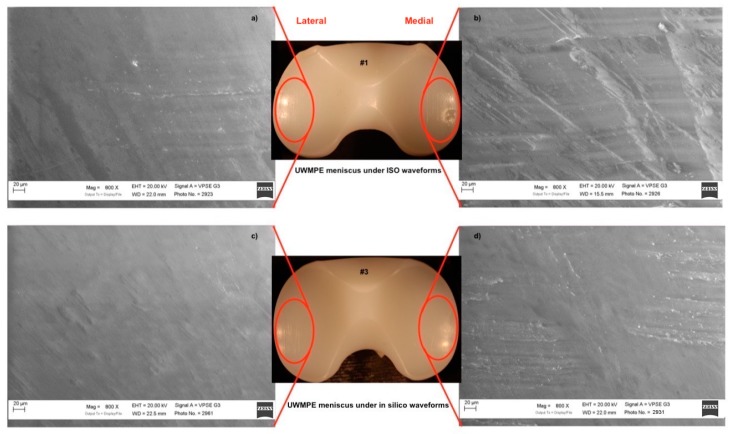
SEM picture on in vitro UHMWPE insert. (**a**,**b**) UHMWPE insert that run under International Organization for Standardization (ISO) waveforms showing a greater variety of damage modes in the focal scars and excessive surface deformation observed deep scratches along the anterior/posterior (AP) direction, pitting, and some crater-like damage. (**c**,**d**) show UHMWPE insert that run under multibody waveforms: scratches were observed along the AP direction.

**Table 1 materials-12-01597-t001:** Geers metric applied for resultant force and moment: in silico vs. in vivo.

	Gait Vaughan	Gait Normal	Gait Fast
Resultant Force	M = −0.11	M = −0.11	M = 0.09
	P = 0.10	P = 0.08	P = 0.10
	C = 0.15	C = 0.13	C = 0.14
Resultant Moment	M = 0.10	M = 0.49	M = 0.92
	P = 0.17	P = 0.09	P = 0.15
	C = 0.20	C = 0.50	C = 0.93

**Table 2 materials-12-01597-t002:** Geers metric applied for force and moment components. in silico vs. in vivo.

	Gait Vaughan	Gait Normal	Gait Fast
Medial/Lateral Force	M = 0.36	M = 8.79	M = 11.18
	P = 0.55	P = 0.58	P = 0.60
	C = 0.66	C = 8.81	C = 11.20
Flexion Moment	M = 0.65	M = 0.61	M = 1.66
	P = 0.19	P = 0.26	P = 0.24
	C = 0.68	C = 0.66	C = 1.68
Anterior/Posterior Force	M = 2.98	M = 3.31	M = 5.38
	P = 0.25	P = 0.25	P = 0.22
	C = 2.99	C = 3.32	C = 5.38
Lateral Moment	M = −0.64	M = 0.47	M = 0.64
	P = 0.18	P = 0.14	P = 0.16
	C = 0.66	C = 0.49	C = 0.66
Proximal/Distal Force	M = −0.14	M = 0.16	M = 0.00
	P = 0.10	P = 0.08	P = 0.10
	C = 0.17	C = 0.18	C = 0.10
Axial Moment	M = 0.21	M = −0.41	M = −0.07
	P = 0.21	P = 0.34	P = 0.48
	C = 0.29	C = 0.53	C = 0.29

**Table 3 materials-12-01597-t003:** Geers metric: in silico vs. ISO 14243.

	Gait Vaughan	Gait Normal	Gait Fast
Flexion/Extension Angle	M = −0.10	M = 0.49	M = 0.92
	P = 0.17	P = 0.09	P = 0.15
	C = 0.20	C = 0.50	C = 0.93
Axial Force	M = −0.07	M = −0.13	M = 0.03
	P = 0.09	P = 0.11	P = 0.10
	C = 0.12	C = 0.17	C = 0.10
Anterior/Posterior Force	M = 2.49	M = 2.60	M = 4.34
	P = 0.32	P = 0.38	P = 0.37
	C = 2.51	C = 2.63	C = 4.35
Rotational Torque	M = 0.43	M = −0.33	M = 0.05
	P = 0.16	P = 0.35	P = 0.47
	C = 0.46	C = 0.48	C = 0.47

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
