# Peer review of "A Critical Analysis of TKR In Vitro Wear Tests Considering Predicted Knee Joint Loads"

_materials, 2019, doi:10.3390/ma12101597_

Round 1
Reviewer 1 Report
I thank the authors for the opportunity to review their interesting work.
In this study authors tried to determine the mechanical force applied to tibial plate for three gait patterns: gait Vaughan, gait normal and gait fast, and compared them to results of ISO14243-1/3. Those results were not in good agreement. The results of in-silico test and wear test were presented; however, the relationship between them were not clear. Authors should analyze location and pattern of wear test and relate them to the in-silico results. Otherwise, readers will not able to find a meaning out of this manuscript.
Minor issues:
Too much introductory part in the abstract.
Line 34-35
Almost the same sentence was repeated.
Page 10
Analyze and elaborate on the results of wear test, compared with in-silico test
Author Response
1. I thank the authors for the opportunity to review their interesting work.
AU: thank you, we appreciate.
2. In this study authors tried to determine the mechanical force applied to tibial plate for three gait patterns: gait Vaughan, gait normal and gait fast, and compared them to results of ISO14243-1/3. Those results were not in good agreement.
AU: The poor agreement between the calculated loads and those from the ISO 14243-1/3 is one of the main interesting results of this research which gave us the aim to investigate the in-vitro wear behavior in TKR accounting for more realistic (in-silico) loads.
In fact the calculated loads are quite different from those reported in the ISO but are in good agreement with those obtained in vivo by Bergman, especially in the case of Vaughan Gait (See Figure 1).
The results of in-silico test and wear test were presented; however, the relationship between them were not clear. Authors should analyze location and pattern of wear test and relate them to the in-silico results. Otherwise, readers will not able to find a meaning out of this manuscript.
AU: thank you very much for this suggestion. We improved the text as suggested.
Minor issues:
Too much introductory part in the abstract.
AU: the abstract was revised entirely.
Line 34-35: Almost the same sentence was repeated.
AU: this sentence was reformulated.
Page 10: Analyze and elaborate on the results of wear test, compared with in-silico test
AU: these results were discussed in the discussion section.
Reviewer 2 Report
The manuscript is focused on in silico approach for the prediction of knee joint load profiles.
The manuscript would benefit an English check: sentences are often too long and report on a latin-based structure. Moreover, "in silico", "in vitro" and "in vivo" should be amended as italic font.
Line 34 "A major limiting factor to the service life of TKRs remains the wear of the polyethylene tibial plate. A major limiting factor to the service life of TKRs remains the wear of the polyethylene meniscus." Please, amend the redundancy.
A TOC graphic/schematic could be extremely helpful to better understand the findings (i.e., represents a comparison among in silico/in vitro/in vivo in case of both successful and unmatching prediction).
Overall, the main criticism relies on the calculated load profiles: in silico studies do not match for all presented situations.
Author Response
The manuscript is focused on in silico approach for the prediction of knee joint load profiles.
1. The manuscript would benefit an English check: sentences are often too long and report on a latin-based structure. Moreover, "in silico", "in vitro" and "in vivo" should be amended as italic font.
AU: the entire manuscript was revised from an English point of view and all the “latin” words were put in italic.
2. Line 34 "A major limiting factor to the service life of TKRs remains the wear of the polyethylene tibial plate. A major limiting factor to the service life of TKRs remains the wear of the polyethylene meniscus." Please, amend the redundancy.
AU: corrected as suggested.
3. A TOC graphic/schematic could be extremely helpful to better understand the findings (i.e., represents a comparison among in silico/in vitro/in vivo in case of both successful and unmatching prediction).
AU: was better rewrote the text.
4. Overall, the main criticism relies on the calculated load profiles: in silico studies do not match for all presented situations.
AU: The poor agreement between the calculated loads and those from the ISO 14243-1/3 is one of the main interesting results of this research which gave us the aim to investigate the in-vitro wear behavior in TKR accounting for more realistic (in-silico) loads.
In fact the calculated loads are quite different from those reported in the ISO but are in good agreement with those obtained in vivo by Bergman, especially in the case of Vaughan Gait (See Figure 1).
Reviewer 3 Report
General Comments:
The significant shortcomings of this paper lie in that it does not reflect the latest and accountable sources of works of literature in the files of predicting in vivo knee loads. Please refer the papers: “Grand Challenge Competition to predict In vivo knee loads” (Fregly, 2012); (Marra 2015); (Carboner, 2015).
Specific Comments:
The title read itself somewhat absurd, “silico knee joint loads”?
Authors said “Direct measurement of joint reaction force is generally possible …”, However, “Grand challenge..” are providing the most comprehensive dataset currently available for subject implanted with a telemetric TKA.
I never heard ISO standard for more accurate TRK in vitro testing. Please verify it.
Author Response
General Comments:
The significant shortcomings of this paper lie in that it does not reflect the latest and accountable sources of works of literature in the files of predicting in vivo knee loads. Please refer the papers: “Grand Challenge Competition to predict In vivo knee loads” (Fregly, 2012); (Marra 2015); (Carboner, 2015).
AU: thank you for these suggestions. The refs were usefully added and included in the Introduction section.
Specific Comments:
1. The title read itself somewhat absurd, “silico knee joint loads”?
AU: The term “in-silico” (Pseudo-Latin for "in silicon", alluding to the mass use of silicon for computer chips) is an expression meaning "performed on computer or via computer simulation" in reference to biological experiments. The phrase was coined in 1989 as an allusion to the Latin phrases in vivo, in vitro, and in situ, which are commonly used in biology (see also systems biology) and refer to experiments done in living organisms, outside living organisms, and where they are found in nature, respectively. https://en.wikipedia.org/wiki/In_silico
However the Authors decided to substitute it by “predicted” for improving the paper readability
2. Authors said “Direct measurement of joint reaction force is generally possible …”, However, “Grand challenge..” are providing the most comprehensive dataset currently available for subject implanted with a telemetric TKA.
AU: thank you for this suggestion however, in our work was impossible to apply a telemetric technique.
3. I never heard ISO standard for more accurate TRK in vitro testing. Please verify it.
AU: the sentence was improved for better reading.
Reviewer 4 Report
I think the paper has been modified from a previous version.
Authors made excellent job. Even if I did not see the previous version, right now the paper is easy to be read and the work is interesting. Moreover the topic fits within the MATERIALS aim and scope.
However some minor concerns should be addressed by the authors for having a more complete paper.
at this stage the manuscript seems too directed to scientes. Please more emphasize the clinical rationale and the clinical implications that could influence the Orhopedic daily practice.
Overall a good paper.
Author Response
1. I think the paper has been modified from a previous version.
AU: thank you.
2. Authors made excellent job. Even if I did not see the previous version, right now the paper is easy to be read and the work is interesting. Moreover the topic fits within the MATERIALS aim and scope.
AU: thank you. We appreciate.
3. However some minor concerns should be addressed by the authors for having a more complete paper.. At this stage the manuscript seems too directed to scientes. Please more emphasize the clinical rationale and the clinical implications that could influence the Orhopedic daily practice.
AU: the manuscript was improved as requested.
5. Overall a good paper.
AU: thank you.
Round 2
Reviewer 1 Report
Authors revised the abstract and added the detailed wear test analysis, followed by added discussion on the analysis. I felt that all the issues I raised were addressed.
Author Response
AU: thank you. We appreciate.
Reviewer 2 Report
As per previous report.
Author Response
1. Comments and Suggestions for Authors
As per previous report.
1. The manuscript would benefit an English check: sentences are often too long and report on a latin-based structure. Moreover, "in silico", "in vitro" and "in vivo" should be amended as italic font.
AU: The manuscript was improved from an English mother tongue and all the suggested words were corrected in Italic.
Reviewer 3 Report
Authors have responded to most of the reviews' comments.